# The effect of arsenic on mitochondrial fatty acid metabolism via inhibition of carnitine palmitoyltransferase 1B and choline kinase beta in C2C12 cells

Fu-Lin Yu[1], Ming-Yan Zhang[2], Gang-Wei Song[1], Yang-Shan Ning[1], Xian Wu[2]*, Yan Gao[1]*

**1** Department of Cardiovascular Medicine, Xi'an Trade Union Hospital, Xi'an, Shaanxi Province, China,
**2** Department of Pharmacology and Toxicology, Brody School of Medicine, East Carolina University, Greenville, North Carolina, United States of America

\* gaoyan_323@163.com (YG); wuxian23@ecu.edu (XW)

## Abstract

Arsenic can enter the human body through environmental exposure via food, drinking water, and chemotherapy for cancer. Prolonged and excessive exposure to arsenic causes various toxic reactions, leading to diseases that significantly impact health and lifespan. Increasing evidence suggests that arsenic damages skeletal muscle tissue by reducing muscle mass and causing atrophy, thereby contributing to conditions such as respiratory and cardiovascular diseases, as well as diabetes. Fatty acid β-oxidation is the most efficient mechanism for ATP production and serves as a primary energy source for tissues, including the heart and skeletal muscles. However, the metabolic mechanisms underlying arsenic's effects on muscle function and pathogenesis remain incompletely understood. In this study, we investigated the role of mitochondrial fatty acid oxidation in arsenic-induced muscular damage using mouse skeletal muscle C2C12 cells. Our results demonstrated a dose-dependent inhibitory effect of sodium arsenite (0–2 μM, 72 hours) on C2C12 cells proliferation, viability, and differentiation (indicated by reduction of myogenic differentiation 1 mRNA expression). Arsenic exposure disrupted mitochondria through increasing reactive oxygen species production, reducing mitochondrial membrane potential to 16.45%, downregulating mitochondrial fatty acid metabolism-related enzymes (carnitine palmitoyltransferase 1B to 15.05% and choline kinase beta mRNA to 49.94%), and decreasing mitochondrial DNA copy number to 42.08%. These findings suggest that arsenic-induced pathological changes in skeletal muscle are associated with impaired mitochondrial membrane function, disrupted fatty acid metabolism, and reduced mitochondrial DNA content in muscle cells.

**Data availability statement:** All relevant data are within the paper and its Supporting information files.

**Funding:** Xian Wu received Pharmacology & Toxicology New Faculty Start-Up Funds in East Carolina University.

**Competing interests:** The authors have declared that no competing interests exist.

## Introduction

Arsenic can enter the food chain through contaminated groundwater or arsenic-rich soil, threatening the health of approximately 200 million people worldwide [1]. According to the WHO 2017 guidelines, the recommended arsenic concentration limit in drinking water is 10 µg/L (0.13 µM) [2]. However, in some areas, groundwater arsenic levels are significantly higher, ranging from 10 ppb to 4 ppm [3]. Arsenic exposure has been linked to severe health consequences, with as many as 1 in 5 deaths attributed to it [4]. In the United States, about 10% of rarely monitored private wells exceed the 10 ppb arsenic limit [5], putting millions at risk of chronic exposure. Even low levels of arsenic pose significant health risks, including the development of various cancers [6] and metabolic syndrome [7]. Prolonged excessive arsenic exposure can result in a range of toxic effects and chronic diseases, such as skin toxicity, neurodegenerative disorders, cardiovascular diseases, and multiple types of tumors [8,9].

Recent epidemiological studies have demonstrated a correlation between arsenic exposure and reduced skeletal muscle mass [10]. Additionally, sodium arsenite exposure (10 mg/kg/day for 4 weeks) has been shown to induce significant cardiotoxic effects in rats, including elevated myocardial tissue enzyme levels, serum creatine kinase MB, and troponin [11]. Despite these findings, the mechanisms underlying arsenic-induced pathological changes in striated muscle, including skeletal and cardiac muscle, remain poorly understood. To explore the potential mechanisms of arsenic-induced skeletal muscle atrophy and loss, we utilized mouse skeletal myoblast C2C12 cells as a model system in this study.

Mitochondria possess their own small chromosome, mitochondrial DNA (mtDNA), which is particularly vulnerable to environmental mutagens and carcinogens. These include free radicals generated by exposure to ionizing radiation, asbestos fibers, and arsenic [12]. Studies have shown that arsenic can directly damage mitochondrial genes in mammalian cells, altering mitochondrial oxidative energy supply and increasing reactive oxygen species (ROS) production [12]. This ROS overproduction can lead to cancer by inducing apoptosis, slowing cell division, and causing cytotoxicity and DNA damage [13]. Mitochondria, as membrane-bound organelles, generate most of the chemical energy required for cellular biochemical reactions. Fatty acids significantly influence mitochondrial energy coupling by increasing the proton conductance of the inner mitochondrial membrane, inhibiting respiration, and triggering the opening of the permeability transition pore [14]. β-oxidation of fatty acids is the most efficient ATP production mechanism and serves as the primary energy source for tissues like the heart and skeletal muscle [15]. Carnitine palmitoyltransferase 1B (CPT1B) is the rate-limiting enzyme in the carnitine shuttle, which facilitates the transport of long-chain fatty acids into mitochondria for β-oxidation. The CPT1B isoform is highly expressed in the heart, skeletal muscle, and brown adipose tissue [16]. In eutherians, the CPT1B gene is closely linked to the choline kinase beta (CHKB) gene, which encodes CHKB, a rate-limiting enzyme in the synthesis of phosphatidylcholine—a major phospholipid in mitochondrial membranes. Recent research has shown that the transcription of CHKB and CPT1B occurs within a shared epigenetic

domain, and their coordinated expression synergistically enhances mitochondrial fatty acid oxidation [17]. This study hypothesizes that sodium arsenite inhibits C2C12 cell proliferation, viability, and differentiation by downregulating CHKB and CPT1B expression, disrupting fatty acid oxidation, and subsequently causing mitochondrial dysfunction.

## Materials and methods

### Cell culture and processing

The C2C12 cell line (ATCC, Manassas, VA, USA) was cultured in DMEM medium (4.5 g/L glucose; Corning Cellgro, Manassas, VA, USA) supplemented with 10% FBS (Corning Cellgro, Manassas, VA, USA) and 1% penicillin-streptomycin (Thermo Fisher Scientific, Greenville, NC, USA). Cells were maintained in a 37°C incubator with 5% $CO_2$. For experiments, C2C12 cells were cultured in 6-, 12-, 24-, or 96-well tissue culture plates and treated (when the confluence reached to around 80%) with varying concentrations of sodium arsenite (0, 0.125, 0.25, 0.5, 1, and 2 μM) [2] for 72 hours.

### Viable cell counting

A hemocytometer was used to count viable cells. Briefly, cells in 24-well plates were digested with 0.53 mM EDTA and 0.05% trypsin (Corning Cellgro, Manassas, VA, USA), then mixed with an equal volume of 0.4% trypan blue solution (Sigma-Aldrich, Raleigh, NC, USA). The number of viable cells was counted under a microscope (n = 4).

### Cell viability

As descript in the reference [18], the MTT assay was performed as follows (n = 8): The original culture medium was removed, and 10 μL of MTT solution (5 mg/mL in PBS) was added to each well of a 96-well plate containing 90 μL of serum-free DMEM. Plates were incubated at 37°C for 3 hours. After incubation, the culture solution was discarded, and 100 μL of dimethyl sulfoxide (Sigma-Aldrich, Raleigh, NC, USA) was added to dissolve the formazan crystals. Plates were protected from light and shaken for 15 minutes before measuring absorbance at 500–600 nm.

### Cell differentiation

Cell differentiation experiments were conducted by culturing cells in differentiation medium, consisting of DMEM (1 g/L glucose; Corning Cellgro, Manassas, VA, USA) with 1% FBS and 1% penicillin/streptomycin, and treated with varying concentrations of sodium arsenite once the cells reached 90% confluency. Cell differentiation was observed daily for 6 days in 6- and 24-well plates (for myogenic differentiation marker MyoD1 mRNA determination, n = 3) under a microscope. On the day 6, cells in the 6-well plates were fixed with 4% paraformaldehyde and stained with hematoxylin (Thermo Fisher Scientific, Greenville, NC, USA) to visualize cell nuclei. Images were captured for analysis.

### Mitochondrial function

JC-1 (Thermo Fisher Scientific, Greenville, NC, USA) is a dye that can penetrate cell membranes and exhibits potential-dependent mitochondrial aggregation properties. In its monomeric form, JC-1 fluoresces green (Ex/Em 515/530 nm), whereas, upon entering the mitochondria, it forms red fluorescent aggregates (Ex/Em 485/590 nm) in a concentration-dependent manner. The red/green fluorescence ratio serves as an indicator of mitochondrial depolarization. The method for detecting mitochondrial function is described in the literature [19]. Briefly, 50 μL of DMEM culture medium containing 10 μg/mL JC-1 was added to each well of a 96-well plate and incubated at 37°C for 45 minutes. The cells were then washed twice with warm PBS. Mitochondrial function was assessed by imaging using a fluorescent microscope or by reading fluorescence at Ex 585/Em 590 and Ex 515/Em 530 using a microplate reader. The red/green fluorescence ratio was calculated to evaluate mitochondrial depolarization (n = 8).

## ROS determination

The ROS detection method is described in the literature [20]. ROS levels in cells cultured in 12-well plates were assessed (n = 3) by measuring the fluorescent signal of DCF (2',7'-dichlorofluorescein), which is produced in the reaction of DCFH-DA (2',7'-dichlorofluorescin diacetate, Sigma-Aldrich, Raleigh, NC, USA) with cell lysate in PBS. Specifically, 10 µL of 150 µM DCFH-DA was added to the cell lysate. The fluorescence of DCF was measured every 5 minutes (Ex 485 nm/Em 530 nm) for 60 minutes at 37°C. ROS levels were normalized using 1 mg of protein in the lysate and a DCF standard curve.

## Amount of mtDNA

DNA extraction was initiated by adding lysis buffer (100 mM NaCl, 10 mM EDTA, 0.5% SDS, and 20 mM Tris-HCl, pH 7.42) and 5 µL of Proteinase K (Sigma-Aldrich, Raleigh, NC, USA; 20 mg/mL) to the cell pellet from a 24-well plate. The mixture was incubated at 55°C overnight, followed by incubation at 37°C for 30 minutes with 5 µL of ribonuclease A (Sigma-Aldrich, Raleigh, NC, USA; 10 mg/mL). DNA was extracted by adding 200 µL of ammonium acetate (7.5 M; Honeywell Aerospace, Rocky Mount, NC, USA) and 500 µL of isopropanol (Sigma-Aldrich, Raleigh, NC, USA).

Quantitative PCR (qPCR) was used to quantify mtDNA, expressed as the ratio of ND1 and CPT1B promoter qPCR products (n = 3).

## mRNA purification, reverse transcription and qRT-PCR

mRNA was extracted from cells in a 24-well plate (n = 3) using the TRIzol method, followed by quantitative reverse transcription PCR (qRT-PCR) to measure the expression of Myod1, CPT1B, and CHKB.

The provided protocol outlines a reverse transcription reaction using the Verso cDNA Synthesis Kit (Thermo Fisher Scientific, Greenville, NC, USA). The reaction, in a final volume of 20 µL, includes specific components for cDNA synthesis: buffer (4 µL), dNTPs (2 µL), RT enhancer (1 µL), primer mix (1 µL), enzyme mix (1 µL), and RNA template (1 µg). The reaction is performed at 42°C for 30 minutes followed by 95°C for 2 minutes.

The qRT-PCR reaction mix included: 1 µL of template cDNA, 1 µL of primer mix (5 µM, primers shown in Table 1), 5 µL of SYBR Green Supermix (Bio-Rad, Hercules, CA, USA), and 3 µL of Milli-Q $H_2O$.

The PCR amplification conditions: 95°C for 5 minutes (initial denaturation), 45 cycles of 95°C for 5 seconds (denaturation), 60°C for 30 seconds (annealing), and 60°C for 1 minute (final extension) (Table 2).

## Statistical analysis

Data analysis was performed using Origin 8.5 software (OriginLab, Northampton, MA) with one-way ANOVA followed by post hoc comparisons using the Holm-Sidak test. All data are presented as the mean ± standard error of the mean (SEM). A p-value of less than 0.05 (p < 0.05) was considered statistically significant.

**Table 1. Primer sequences for mitochondria DNA quantification by qRT-PCR.**

|  | Forward primer | Reverse primer |
| --- | --- | --- |
| ND1 | CTAGCAGAAACAAACCGGGC | CCGGCTGCGTATTCTACGTT |
| CPT1B promotor | CTCCTGGTGACCTTTTCCCT | CACAAGGTTGCTGGAAGGTC |

**Table 2. Primer sequences for mRNA quantification by qRT-PCR.**

| Transcript | Forward primer | Reverse primer |
| --- | --- | --- |
| B2M | CCCTGGTCTTTCTGGYGCTT | CGTAGCAGTTCAGTAYGTTCG |
| Myod1 | GACACCGCCTACTACAGTGA | CACTATGVTGGACAGGCAGT |
| CPT1B | CAACTCCTGGAAGAAACGCC | TCCACCTTGCAGTAGTTGGA |
| CHKB | TCCCTGAGATGAACCTGCTG | GGCGATGGGGTAGACTCTAG |

## Results

### Sodium arsenite inhibited C2C12 cell proliferation and activity

To evaluate the effect of sodium arsenite on cell proliferation, the number of cells was counted in each group after 72 hours of treatment. We found that the cell counts in the 2 µM sodium arsenite group was significantly decreased (32.52%, $p < 0.05$) compared to the control group (Fig 1A). As an indicator of cell viability, proliferation, and cytotoxicity, the MTT assay results showed that the viability of C2C12 myoblasts was significantly inhibited (55.62% at 2 µM, $p < 0.05$) at concentrations starting from 1 µM of sodium arsenite (Fig 1B).

### Sodium arsenite inhibited C2C12 cell differentiation

C2C12 cells differentiate into myotubes, a muscle phenotype, under specific metabolic conditions (differentiation medium). In our study, on Days 1 and 2, all cells remained in the myoblast stage (Fig 2A). On Day 3, cells in the control group began to differentiate into long myotubes (Fig 2B). Myoblasts in the 0.125 µM sodium arsenite-treated group began to differentiate on Day 4, while cells in the 0.25 µM sodium arsenite-treated group began to differentiate on Day 5. Cells in the 0.5 µM, 1 µM, and 2 µM sodium arsenite-treated groups began to differentiate on Day 6.

To quantify myogenic differentiation, the expression of Myod1 (a myogenic regulatory transcription factor) was measured by qRT-PCR on Day 6. The results showed that the Myod1 mRNA level in cells cultured in differentiation medium (control for sodium arsenite) was significantly higher ($p < 0.05$) compared to myoblasts cultured in normal medium (Fig 2C). Additionally, treatment with 0.5 µM, 1 µM, and 2 µM sodium arsenite dose-dependently inhibited (60.49% at 2 µM, $p < 0.05$) Myod1 mRNA expression (Fig 2C).

### Sodium arsenite inhibited mitochondrial membrane potential

To determine the role of mitochondria in the inhibition of proliferation, viability, and differentiation of C2C12 cells by sodium arsenite, a mitochondrial membrane potential-dependent fluorescent probe, JC-1, was used in the present study. Fig 3A and 3B show the intracellular monomeric form of JC-1 (green fluorescence) and the aggregate form (red fluorescence) after entering the mitochondria in C2C12 cells, respectively.

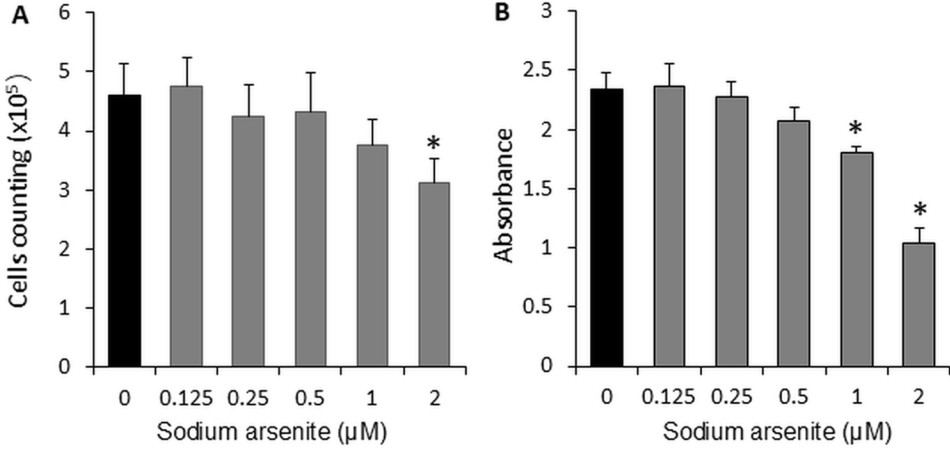

**Fig 1. Sodium arsenite inhibited C2C12 cells proliferation (A) and viability (B).** Cells were cultured in 24-well or 96-well plates and treated with graded concentrations (0–2 µM) of sodium arsenite for 72 hours. Data are presented as mean±SEM. *$p < 0.05$ compared with the control group (0 µM sodium arsenite).

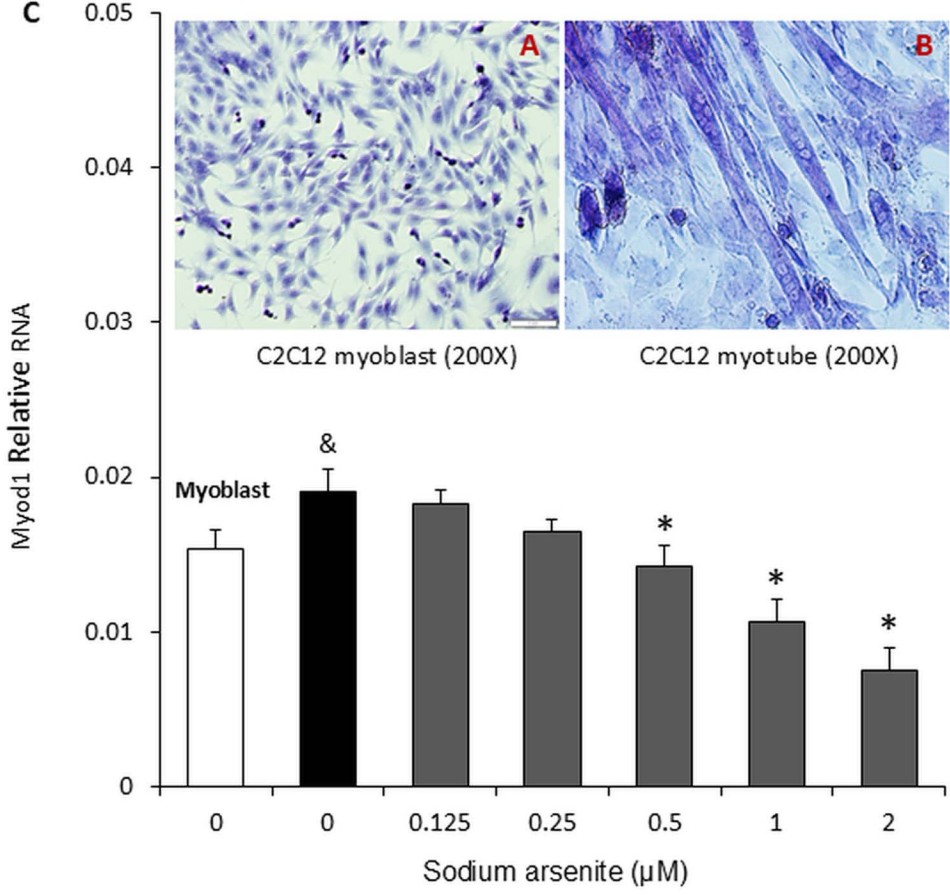

**Fig 2. Sodium arsenite inhibited the differentiation of C2C12 cells.** C2C12 cells were cultured in 24-well plates, and upon reaching 90% confluency, the growth medium was replaced with differentiation medium (DMEM, 1% FBS, 1 g/L glucose) containing graded concentrations of sodium arsenite (0–2 µM). Images A and B show C2C12 myoblasts and differentiated mature myotube cells, respectively. The bar graph (C) illustrates Myod1 RNA levels on the 6th day of differentiation. Data are presented as mean ± SEM. &p < 0.05 compared with the myoblasts; *p < 0.05 compared with the control group (0 µM sodium arsenite).

The fluorometric assay data revealed that compared to the control group, 0.05 pM sodium arsenite had no significant effect on mitochondrial membrane potential. However, treatment with 0.26 pM sodium arsenite significantly impacted (p < 0.05) the mitochondrial membrane potential of C2C12 cells after 72 hours, reducing it by approximately 83.65%. Interestingly, even increasing the sodium arsenite concentration to 2 µM resulted in a similar reduction in mitochondrial membrane potential as observed with the 0.26 µM treatment (Fig 3C).

## Sodium arsenite increased ROS production

Our data showed that even at a very low concentration of sodium arsenite (0.26 pM), mitochondrial membrane function was significantly reduced. It is well known that the major source of endogenous ROS in cells is mitochondrial oxidative phosphorylation. To assess mitochondrial metabolic function, we measured ROS production in C2C12 cells using the DCFH probe, which detects ROS. The data indicated that treatment with sodium arsenite for 72 hours increased (1.63-fold at 2 µM, p < 0.05) ROS production in a dose-dependent manner, ranging from 0.125 µM to 2 µM (Fig 4).

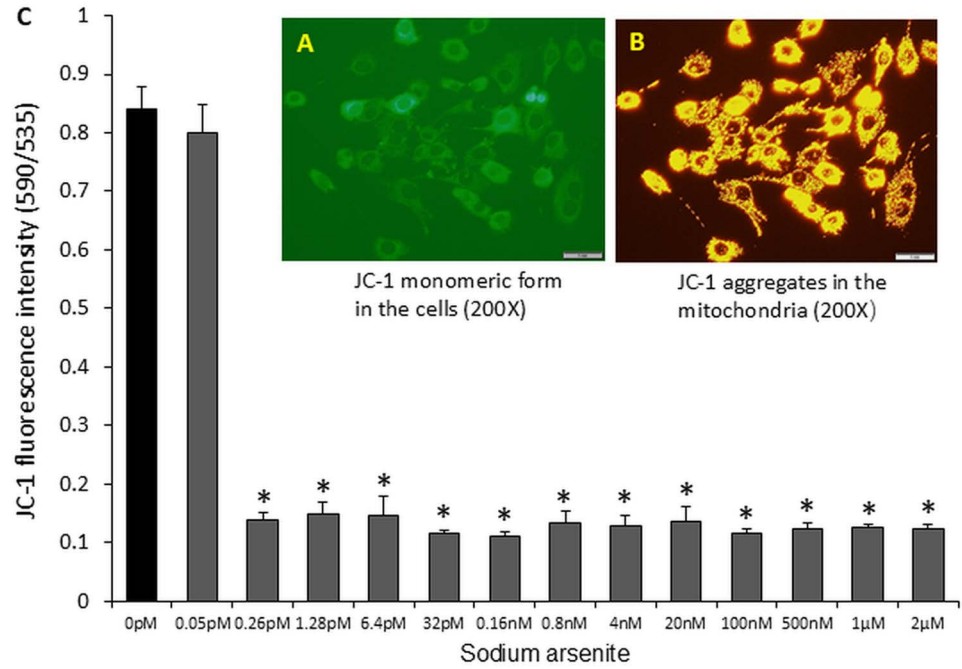

**Fig 3. Sodium arsenite impaired mitochondrial membrane function (membrane potential) in C2C12 cells.** Cells were cultured in 96-well plates and treated with graded concentrations of sodium arsenite (0–2 μM) for 72 hours. Images A and B show intracellular haplotype and intramitochondrial aggregated JC-1, respectively. The bar graph (C) represents the ratio of red fluorescent aggregates (indicative of intact mitochondria) to green fluorescence (monomeric form, indicative of depolarized mitochondria), reflecting the degree of mitochondrial depolarization. Data are presented as mean ± SEM. *p < 0.05 compared with the control group (0 μM sodium arsenite).

### Sodium arsenite inhibited mtDNA

The increase in ROS production may cause damage to mtDNA. Our data suggest that mitochondria play a key role in the inhibition of C2C12 cells proliferation, viability, and differentiation by sodium arsenite. In this part of the study, we examined whether sodium arsenite affects the amount of mtDNA in C2C12 cells. The results showed that treatment with 0.625 μM sodium arsenite had no significant effect on the mtDNA content in C2C12 cells after 72 hours. However, concentrations ranging from 0.25 μM to 2 μM of sodium arsenite dose-dependently reduced (57.92% at 2 μM, p < 0.05) the amount of mtDNA (Fig 5).

### Sodium arsenite inhibited CPT1B and CHKB mRNA expression

Arsenite primarily targets mitochondria, disrupting their function and interfering with the process of fatty acid β-oxidation. In this study, we examined the effect of sodium arsenite on the rate-limiting enzymes involved in fatty acid oxidation and phosphatidylcholine synthesis, specifically CPT1B (Fig 6A) and CHKB (Fig 6B). The results showed that sodium arsenite significantly inhibited the mRNA expression of both CPT1B (84.95% at 2 μM, p < 0.05) and CHKB (50.06% at 2 μM, p < 0.05) in C2C12 cells at concentrations ranging from 0.5 μM to 2 μM.

### Discussion

Our data in the present study demonstrate that sodium arsenite exhibits a dose-dependent inhibition of proliferation and differentiation in C2C12 cells. It is well established that arsenite primarily targets mitochondria, disrupts their function, and interferes with the process of fatty acid β-oxidation. In this study, it is we found that sodium arsenite could

 

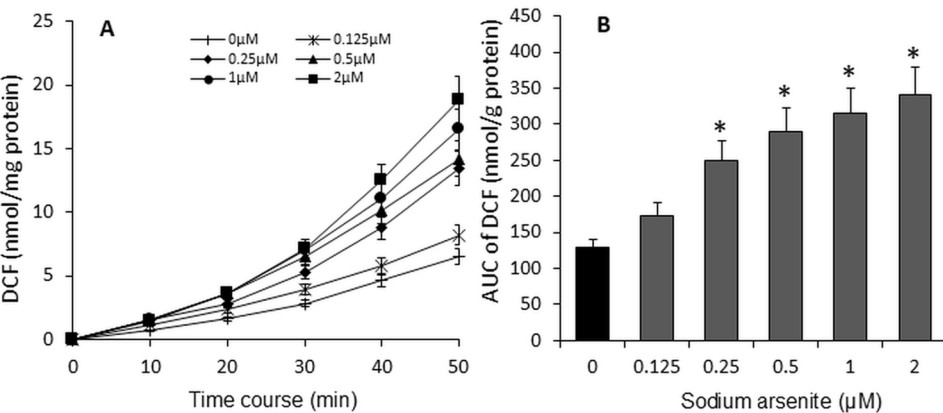

**Fig 4. Sodium arsenite dose-dependently increased ROS production in C2C12 cells.** Cells were cultured in 12-well plates and treated with graded concentrations of sodium arsenite (0–2 µM) for 72 hours. The line graph (A) illustrates the amount of DCF produced over time, resulting from the reaction between ROS and DCFH. The bar graph (B) represents the area under the curve (AUC) of DCF production over 50 minutes. Data are presented as mean±SEM. *p<0.05 compared with the control group (0 µM sodium arsenite).

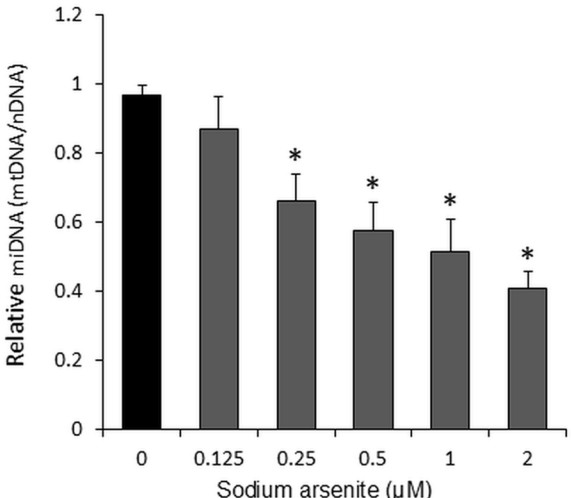

**Fig 5. Sodium arsenite dose-dependently inhibited mtDNA replication or induced DNA damage in C2C12 cells.** Cells were cultured in 24-well plates and treated with graded concentrations of sodium arsenite (0–2 µM) for 72 hours. The amount of mtDNA is expressed as the ratio of mitochondrial DNA (mtDNA) to nuclear DNA (nDNA). Data are presented as mean±SEM. *p<0.05 compared with the control group (0 µM sodium arsenite).

reduce mitochondrial membrane potential by eightfold compared (Fig 3). Furthermore, it delayed cell differentiation. and increased the production of ROS (Fig 4), decreased mitochondrial DNA content (Fig 5), inhibited the expression of Myod1 (Fig 2), CPT1B, and CHKB (Fig 6) RNA. Additionally, cell viability and proliferation were inhibited at 1 µM and 2 µM, respectively (Fig 1), which might indicate the differentiation is more sensitive and should be the target time window for therapy or prevention of arsenic exposure. All these results suggest that mitochondrial fatty acid metabolism, including fatty acid oxidation and the synthesis of phosphatidylcholine, plays a crucial role in the inhibitory effects of sodium arsenite on C2C12 cells proliferation and differentiation.

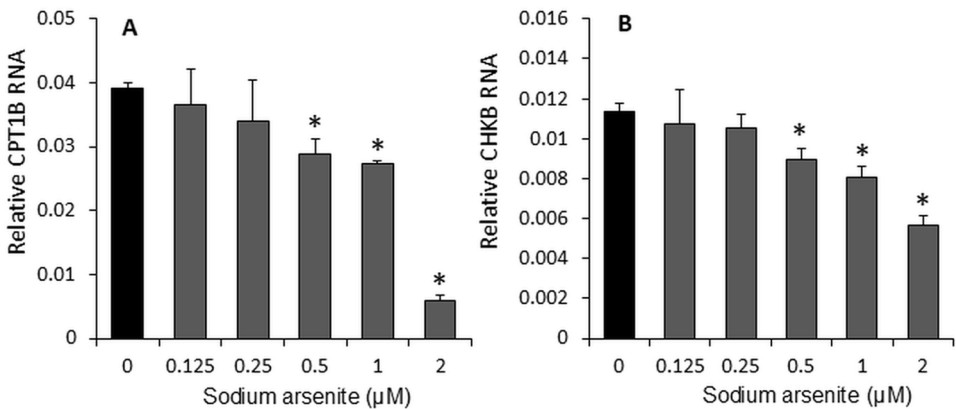

**Fig 6. Sodium arsenite dose-dependently inhibited the RNA expression of CPT1B (A), a rate-limiting enzyme of the carnitine shuttle for fatty acid β-oxidation, and CHKB (B), a rate-limiting enzyme in the synthesis of phosphatidylcholine, in C2C12 cells.** Cells were cultured in 24-well plates and treated with graded concentrations of sodium arsenite (0–2 μM) for 72 hours. Data are presented as mean ± SEM. *p < 0.05 compared with the control group (0 μM sodium arsenite).

The U.S. Food and Drug Administration has recommended sodium arsenite for the treatment of malignant tumors [21]. However, after treatment, low doses of sodium arsenite can remain in the blood and body for extended periods, potentially impacting cellular health. Studies have demonstrated the toxicity of sodium arsenite on embryonic primary rat brain neuroepithelial cells and shown that long-term residual sodium arsenite at nanomolar concentrations can affect the activity of bone marrow mesenchymal stem cells [22]. Our data indicate that even a very low dose of sodium arsenite (0.26 pM) administered over 72 hours significantly reduced the mitochondrial membrane potential in C2C12 cells (Fig 3). This reduction could impair energy production, disrupt normal cell signaling pathways, and ultimately cause physiological damage to cells, tissues, organs, and the entire body over time.

Studies have shown that sodium arsenite induces the inhibition of cell proliferation in activated mouse T cells [23] and suppresses the proliferation of muscle tissue in rodent models [24]. In our study, we observed that sodium arsenite inhibited proliferation in C2C12 cells at a concentration of 2 μM (Fig 1A), and 1 μM of sodium arsenite significantly reduced cell viability (Fig 1B). These findings align with previous experiments, which reported that sodium arsenite decreased the viability of human and rat mesenchymal stem cells [25,26]. Additionally, it has been observed that sodium arsenite at lower concentrations (0–0.1 μM) and longer exposure durations (5–21 days) causes a dose- and time-dependent reduction in cell viability [22].

The differentiation of skeletal muscle is a highly regulated, multistep process. Initially, single muscle cells proliferate freely, then align and fuse to form multinucleated myotubes. In vivo, this process is controlled by the complex interplay of various growth and trophic factors [27]. At very low doses of arsenic (100 ppb or less in drinking water) [28] and early postnatal exposure has been shown to induce abnormal muscle actin. Sodium arsenite has been found to inhibit muscle differentiation in female Killifish, leading to an increased likelihood of producing juveniles with curved bodies—a condition linked to alterations in myosin light chains [29]. Similarly, arsenic exposure has been reported to cause comparable changes in cardiovascular smooth muscle [30]. Mouse C2C12 myoblasts can differentiate into immature muscle tube cells under low-glucose and low-serum conditions [31]. Previous studies have shown that 20 nM sodium arsenite reduces the expression of the transcription factor myogenin, thereby delaying C2C12 cell differentiation [32,33]. In our experiment, a slightly higher dose of sodium arsenite (0.125 μM) delayed differentiation and inhibited Myod1 RNA expression in C2C12 cells. This difference may be attributable to variations in culture conditions.

Mitochondria play a central role in cellular biological activities and are a significant source of ROS. The respiratory chain is closely associated with ROS production [34,35]. ROS can directly act as mutagens or indirectly function as

messengers and regulators, impacting all structural and functional components of cells and various aspects of cell biology. Excessive ROS not only cause genomic mutations but also induce irreversible oxidative modifications of proteins, lipids, and glycans, impairing their function and contributing to diseases or cell death [36]. The pathological effects of ROS are extensive, including oxidative damage to cells by sequestering electrons from lipids in phospholipid membranes [37] and interfering with cell signaling [38]. Arsenic has been linked to mitochondrial dysfunction through several mechanisms, including elevated ROS production. Studies have identified increased oxidative stress as a primary cause of arsenite toxicity, with its effects being dose-dependently. The doses (0.5–5 µM) of sodium arsenite have been shown to elevate ROS production and cause mitochondrial membrane depolarization in human breast cancer MCF-7 cells [39]. Our findings align with these observations, demonstrating that sodium arsenite dose-dependently (0.125–2 µM) increased ROS production in C2C12 cells (Fig 4).

The mitochondrial membrane potential generated by proton pumps is an essential component of energy storage during oxidative phosphorylation and is critical for mitochondrial functions. Arsenite can enter mitochondria via aquaglyceroporins, where it binds and inhibits several enzymes involved in energy production, where it can bind and inhibit numerous enzymes involved in energy production [40], including complexes II and IV of the electron transport chain [41]. A self-amplifying loop of mitochondrial dysfunction has been identified, where mitochondrially derived ROS activate ERK1/2, which then translocates to the mitochondria. This cascade damages ATP synthase function, reduces mitochondrial membrane potential, and causes cytochrome c release, leading to further ROS production [42]. Studies have also shown that arsenic decreases membrane-based enzyme activity, such as Na+/K+ATPase, through ROS generation, disrupting membrane homeostasis critical for maintaining mitochondrial transmembrane potential [43]. Additionally, arsenic-induced toxicity mechanisms include loss of mitochondrial membrane potential, ROS generation, and lipid peroxidation [44]. In our study, sodium arsenite reduced mitochondrial membrane potential by approximately eightfold, (Fig 3). Fatty acid β-oxidation in mitochondria is the most efficient mechanism for ATP production and serves as the primary energy source for heart and skeletal muscle tissues [15]. CPT1B, a rate-limiting enzyme in the carnitine shuttle for fatty acid β-oxidation, is highly expressed in heart and muscle tissues [16]. CPT1b deficiency can cause lipotoxicity in the heart under pathological stress, leading to exacerbation of cardiac pathology [45]. In our study, CPT1B expression was significantly suppressed by sodium arsenite. Similarly, CHKB, a rate-limiting enzyme in the synthesis of phosphatidylcholine, a predominant mitochondrial membrane phospholipid, play a key role to maintain the mitochondrial membrane stability [46], was inhibited sodium arsenite simultaneously. This finding aligns with recent research showing that the transcription of the CHKB and CPT1B genes is coordinated within a unitary epigenetic domain, and their expression synergistically enhances mitochondrial fatty acid oxidation [17] and the human disease caused by disruption of a phospholipid de novo biosynthetic pathway, demonstrating the pivotal role of phosphatidylcholine in muscle [47].

An individual cell contains hundreds of mitochondria, each housing multiple copies of the mitochondrial genome [48]. Unlike nuclear DNA, mtDNA lacks robust protection and repair mechanisms and exists in an environment of high oxidative stress. Consequently, the mutation rate of mtDNA is significantly higher than that of nDNA [48]. By calculating the mtDNA-to-nDNA ratio and measuring relative changes in mtDNA copy number, it is possible to infer cell function and the extent of cellular damage [49]. ROS play a crucial role in DNA damage [39], and arsenic-induced genotoxic effects have been shown to be ROS-dependent [12]. Conversely, a reduction in mtDNA copy number could impair mitochondrial function, potentially increasing cellular ROS levels [12]. This feedback loop—where increased ROS levels induced by arsenic exposure raise the mutation rate and further reduce mtDNA quantity—can disrupt cellular and tissue functions, ultimately harming health. In our study, we observed that sodium arsenite-induced ROS production (Fig 4) was accompanied by a dose-dependent (0.125–2 µM) decrease in mtDNA copy number (Fig 5).

Arsenic exposure has been shown to reduce skeletal muscle mass [10] and cause cardiotoxicity [11]. Mitochondria, which play a central role in cellular biological activities by producing ATP as the main energy source, rely on fatty acid β-oxidation as the most efficient energy production mechanism. In this study, we used mouse skeletal muscle C2C12

cells as a model to investigate the role of mitochondria and fatty acid β-oxidation in arsenic-induced skeletal damage and cardiotoxicity. Our findings suggest that mitochondrial function and fatty acid β-oxidation are critical factors in the arsenic-induced inhibition of skeletal muscle cell proliferation and differentiation. These processes are essential for skeletal muscle regeneration, a natural mechanism of muscle repair and renewal. Although our data is based on the mouse cell line, the finding of this study provide a clue to understand the mechanisms of arsenite on human skeletal muscle damages, which is the limitation of this study, but in the future, we will address it on human pluripotent stem cells induced skeletal muscle cells and cardiomyocytes and its organoid.

## Supporting information

**S1 Data. Minimal data set.**
(XLSX)

## Acknowledgments

We thank Dr. Fanrong Yao for technical support.

## Author contributions

**Conceptualization:** Xian Wu, Yan Gao.

**Data curation:** Fu-Lin Yu.

**Formal analysis:** Ming-Yan Zhang, Gang-Wei Song.

**Investigation:** Gang-Wei Song.

**Methodology:** Ming-Yan Zhang, Yang-Shan Ning.

**Writing – original draft:** Fu-Lin Yu.

**Writing – review & editing:** Xian Wu, Yan Gao.

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
