## [Decision Letter · Decision Letter 0]

9 Apr 2025

PONE-D-25-09161The Effect of Arsenic on Mitochondrial Fatty Acid Metabolism via Inhibition of Carnitine Palmitoyltransferase 1B and Choline Kinase Beta in C2C12 CellsPLOS ONE

Dear Dr. Wu,

Thank you for submitting your manuscript to PLOS ONE. After careful consideration, we feel that it has merit but does not fully meet PLOS ONE’s publication criteria as it currently stands. Therefore, we invite you to submit a revised version of the manuscript that addresses the points raised during the review process.

We look forward to receiving your revised manuscript.

Kind regards,

Muhammad Zubair

Academic Editor

PLOS ONE

Journal Requirements:

Reviewers' comments:

Reviewer's Responses to Questions

**Comments to the Author**

1. Is the manuscript technically sound, and do the data support the conclusions?

Reviewer #1: Yes

2. Has the statistical analysis been performed appropriately and rigorously? 

Reviewer #1: I Don't Know

3. Have the authors made all data underlying the findings in their manuscript fully available?

Reviewer #1: No

4. Is the manuscript presented in an intelligible fashion and written in standard English?

Reviewer #1: Yes

5. Review Comments to the Author

Reviewer #1: The study makes a valuable contribution to the understanding of arsenic-induced muscle toxicity and is suitable for publication after minor revisions. Here are a few suggestions before proceed to publishing:

Abstract is effectively summarizing the study which addresses a relevant mechanism of arsenic toxicity in skeletal muscle.

1. Mentioning the used analytical methods maybe better for scientific rigor.

2. Including a few representative findings such as % decrease in membrane potential, or fold change would add impact.

Introduction is well structured and well referenced. It builds a brief scientific rationale.

3. The aim of the study statement is given, however, the paper would benefit from a hypothesis statement to give a rather clear and mechanistic model for the study.

Materials and methods:

4. Seeding density for each test is important and should be given.

5. The sample size, biological and technical replicates were not mentioned in none of the experiments, and should be given

6. Reverse transcription of mRNA was not included. It is better to be included, in terms of mentioning the initial mRNA amount for following qPCR analysis.

Results:

7. Phrases like “significantly impacted” or “significantly inhibited” (e.g., Lines 170, 172, 184, 222) should be accompanied by specific p-values and a statistical method reference. This could be given in the manuscript or in the figure legends.

8. The results mention decreases or inhibition, but the magnitude of changes is often not described.

Discussion:

9. Condensing some sections which are already given in the results (e.g. line 227- 237, lines 303 – 309 ..)

10. Given the relevance to human exposure and environmental health, it would be useful to include a brief comment on how these findings could be translated.

11. How decreased fatty acid biosynthesis related to CHKB suppression might impact mitochondrial membrane stability can be further discussed.

12. At the very end of the last paragraph, future directions would better be given.

The manuscript is intelligible and written in standard English, but would benefit from minor language editing to enhance clarity and polish the overall presentation. Some of the occasional awkward phrasings and punctuation errors are stand out.

6. PLOS authors have the option to publish the peer review history of their article (what does this mean? ). If published, this will include your full peer review and any attached files.

**Do you want your identity to be public for this peer review?** For information about this choice, including consent withdrawal, please see our Privacy Policy .

Reviewer #1: No

---

## [Author Response · Author response to Decision Letter 1]

24 Apr 2025

Response to reviewers and editors are uploaded in the attach files

---

## [Editor Report · Decision Letter 1]

30 Apr 2025

The Effect of Arsenic on Mitochondrial Fatty Acid Metabolism via Inhibition of Carnitine Palmitoyltransferase 1B and Choline Kinase Beta in C2C12 Cells

PONE-D-25-09161R1

Dear Dr. Wu,

We’re pleased to inform you that your manuscript has been judged scientifically suitable for publication and will be formally accepted for publication once it meets all outstanding technical requirements.

Kind regards,

Muhammad Zubair

Academic Editor

PLOS ONE
---

## [Editor Report · Acceptance letter]

PONE-D-25-09161R1

PLOS ONE

Dear Dr. Wu,

I'm pleased to inform you that your manuscript has been deemed suitable for publication in PLOS ONE. Congratulations! Your manuscript is now being handed over to our production team.

Kind regards,

on behalf of

Dr. Muhammad Zubair

Academic Editor

PLOS ONE